# ARE LARGE VISION-LANGUAGE MODELS GOOD ANNOTATORS FOR IMAGE TAGGING?

## ABSTRACT

Image tagging, a fundamental vision task, traditionally relies on human-annotated datasets to train multi-label classifiers, which incurs significant labor and costs, especially for large-scale label spaces. While Large Vision-Language Models (LVLMs) offer promising potential to automate annotation, their capability to replace human annotators remains underexplored. This paper systematically investigates LVLMs' annotation quality and the performance of models trained on LVLM-generated labels. Our analysis reveals that LVLMs achieve competitive performance on common categories but lag behind humans on uncommon or ambiguous categories. Surprisingly, models trained on LVLM-generated labels outperform those trained on human-annotated labels in certain categories, suggesting imperfections in human annotations. Motivated by these findings, we propose LVLMANT, a novel framework for image tagging, which aims to achieve human-level annotation ability. LVLMANT comprises two components: Prompts-to-Candidates (P2C), which employs group-wise prompting and annotation ensembling to efficiently produce a candidate set that covers as many true labels as possible while reducing subsequent annotation workload; and Concept-Aligned Disambiguation (CAD), which interactively calibrates the semantic concept of categories in the prompts and effectively refines the candidate labels. Extensive experiments on benchmark datasets demonstrate LVLMANT's effectiveness in balancing annotation quality and automation, significantly reducing reliance on manual effort while achieving performance comparable to human annotations.

## 1 INTRODUCTION

Image tagging (Chen et al., 2019; Zhang et al., 2024) is a fundamental and practical vision task that aims to annotate an image with all its relevant labels. Typical methods follow the multi-label learning paradigm (Liu et al., 2021), which involves training a multi-label classifier on a human-annotated dataset and then using it to predict all relevant labels for unseen images. While this framework has achieved remarkable success, it suffers from a potential limitation: The requirement of manually annotating a new dataset for each new task. This annotation process can be particularly labor-intensive and costly, especially in scenarios with a large label space with hundreds or even thousands of categories. Therefore, it is crucial to reduce the cost of manual annotation for achieving efficient image classification.

Large Vision-Language Models (LVLMs) (Radford et al., 2021; Yang et al., 2023) have emerged as a transformative technology with the potential to reshape the traditional supervised learning paradigm. Trained on massive cleaned web data and synthesized data, these advanced models exhibit powerful multimodal understanding capabilities across diverse domains, and in some cases, their performance can even rival that of humans. A key question is *whether LVLMs are good annotators for image tagging*. Can they produce human-level annotations and, under appropriate conditions, replace or at least complement human annotators?

To answer this question, we conduct a systematic study to investigate the ability of LVLMs on image tagging. On the one hand, we categorize prompts into two types, *multi-option* prompting, which asks what objects are present in each image and restricts outputs to a predefined candidate set, and *binary* prompting, which asks whether a specific category is present and iterates over all categories. The former has low annotation cost (one inference per image) and achieves high recall but low pre-

cision. The latter has high annotation cost (it requires $q$ inferences per image, where $q$ is the number of categories) and achieves high recall and precision. These findings provide a guidance for the design of our proposed annotation framework. On the other hand, we evaluate the annotation ability of LVLMs from two perspectives: the quality of LVLM-generated annotations and the performance of models trained on these annotations. For annotation quality, we find that LVLMs show promising performance, but still exhibit a noticeable gap compared to human annotators. The reason is intuitive: they perform well on common categories, whereas their performance significantly degrades on uncommon or ambiguous ones. Regarding downstream model performance, even when some labels are missing compared to human annotations, training models using only the true positives (according to human annotations) from LVLM-generated labels leads to only a minor degradation in performance. For some categories, models trained on LVLM-generated annotations even outperform those trained on human-annotated data. These findings indicate that human annotations can be imperfect, and LVLM-generated annotations offer the dual benefit of substantially reducing manual annotation costs while potentially enhancing annotation quality.

Based on these observations, we introduce an LVLM-driven annotation framework, LVLMANT, which aims to produce human-level annotations for image tagging. Our framework is two staged: a *multi-option* prompting stage that efficiently produces a compact candidate pool, typically comprising fewer than one tenth of the categories, followed by a *binary* prompting verification stage that carefully refines the candidates and significantly enhances annotation quality. This structured annotation framework also benefits from the complementary nature of multi-option prompting and binary prompting, as the two annotation strategies exhibit different annotation behaviors. As a result, it achieves significantly higher annotation quality than using either strategy alone. Specifically, in the first stage, we adopt a group-wise prompting strategy that clusters categories which rarely co-occur into the same group to reduce within-group competition and integrates an ensemble of LVLMs to increase annotation recall. Empirically, we find that many false positives in candidate sets are caused by what we refer to as *concept misalignment*, where the category name does not accurately correspond to the actual object concept. To address this issue, in the second stage, we introduce a concept-aligned disambiguation method, which leverages ChatGPT-4o to interactively calibrate category names for refining candidate label sets. By integrating the complementary strengths of two prompting methods, LVLMANT delivers efficient operation and high-quality annotations. Importantly, every stage substantially shrinks the candidate label sets, enabling us to introduce human-assisted calibration at low annotation cost and achieve human-LVLM collaborative annotation, which in turn significantly enhances annotation quality. Extensive experimental results on multiple benchmark datasets validate the effectiveness of the proposed method.

## 2 A CLOSE LOOK TO LVLM-GENERATED ANNOTATIONS

In this section, we present a systematic investigation of LVLM-based annotation for the image-tagging task, with the goal of characterizing how it differs from human-provided labels. We begin by introducing notation and settings.

### 2.1 NOTATION AND SETTINGS

We are given a dataset $\{x_i\}_{i=1}^{n}$ consisting of $n$ images, where $x_i \in \mathcal{X}$ and $\mathcal{X} = \mathbb{R}^d$ is the feature space, along with a category vocabulary $\{C_1, C_2, \ldots, C_q\}$ of $q$ category names to be annotated. Each image is associated with an unknown label vector $y_i \subseteq \mathcal{Y}$, where $\mathcal{Y} = \{0,1\}^q$ is the label space with $q$ possible labels. Here, $y_k = 1$ indicates that the $k$-th label with the name $C_k$ is relevant while $y_k = 0$ indicates that it is not. We use $[q]$ to denote the integer set $\{1, 2, \ldots, q\}$.

To disclose the characteristics of LVLM-generated annotations, we employ one of the most powerful open-source LVLMs, Qwen2.5-VL, to annotate two widely used multi-label image benchmark datasets, MS-COCO 2014 (COCO 2014) (Lin et al., 2014) and Objects365 (O365) (Shao et al., 2019). Given the flexibility of natural language, the number of possible prompt templates is virtually unlimited. In our study, we categorize prompt templates into three types: *open-ended*, *multi-option* and *binary*. *Open-ended* templates ask large vision-language models (LVLMs) what objects are present without imposing constraints, *e.g.*, What objects are in this image?. *Multi-option* templates pose the same question but restrict the answer to a predefined candidate set, *e.g.*, What objects are in this image? Candidates:<object list>.. *Bi-*

Figure 1: An example of LVLM-generated annotations using different prompting methods. Green denotes true labels; Red denotes missed labels.

*nary* templates differ from both by querying the presence of a specific object, *e.g.*, Is there a <object> in this image?", to which the LVLM needs only answer "Yes" or "No".

## 2.2 PROMPT MATTERS: OPEN-ENDED VS. MULTI-OPTION VS. BINARY

To investigate how prompt templates affect generated annotations, we evaluate Qwen2.5-VL under different templates and present the generated labels in Figure 1. From the figure, we observe: (i) *open-ended* prompts are suboptimal for image tagging because the model often produce labels outside the candidate set, frequently at a higher semantic level (*e.g.*, *food*, *utensils*). While these labels are not strictly incorrect, they provide limited value for downstream model training. (ii) *multi-option* and *binary* prompts are better suited to the task but exhibit different behaviors: *multi-option* prompting tends to output more labels, covering most ground-truth ones yet introducing noise, whereas binary prompting prioritizes precision and may miss some true labels. This difference mainly stems from how the LVLM handle the two prompting methods. For *multi-option* prompting, the model generates labels sequentially and later tokens depend on earlier ones. When visual evidence is weak (for example, due to occlusion or very small objects), the model tends to rely on textual co-occurrence and produces additional labels, which yields broader coverage of ground-truth labels, *e.g.*, *cup* in the example. For *binary* prompting, the model makes an independent Yes/No decision for a single class without conditioning on other information, so it predicts only when visual evidence is sufficient and often misses weak positives, *e.g.*, *bowl* in the example.

To further investigate these two prompting methods, we use Qwen2.5-VL and InternVL3 to annotate two datasets. Figure 2(a) and Figure 2(b) show annotation quality in terms of per-Class Precision (CP) and per-Class Recall (CR) scores. It can be observed that *multi-option* prompting typically yields high recall but low precision, whereas *binary* prompting achieves relatively high precision and recall, although their recall remains lower than that of *multi-option* prompts. These results provide guidance for designing strategies to improve annotation quality.

## 2.3 COMPARISON WITH HUMAN ANNOTATIONS

Figure 2 and Figure 3 illustrate the quality of annotations generated by Qwen2.5-VL and the performance of models trained on these LVLM-generated annotations. Based on these results, we summarize three empirical findings.

**Finding 1: LVLM-generated annotations exhibit a significant disparity between common and uncommon/ambiguous categories.** Figure 2(a) and Figure 2(b) illustrate the annotation quality in terms of CP and CR scores. By using binary prompts (the red bars), Qwen2.5-VL produces high-quality annotations on COCO 2014, which primarily consists of common object categories [1]. In contrast, on O365, which includes a large number of ambiguous categories, the quality of the generated labels drops significantly. To provide a further validation for this phenomenon, Figure 2(c) illustrates the annotation quality for the top-10 and bottom-10 categories in terms of F1 scores. It can be observed that the top-10 categories are mostly common objects, such as *person* and *cat*, or conceptually unambiguous categories like *guitar*, *golf club*, *baseball bat*; while the bottom-10 categories tend to be ambiguous categories, *e.g.*, *mask*, *tape*, and *marker*. The quality of annotations has a direct impact on model performance. As shown in Figure 3(a), when trained on LVLM-annotated data, the model achieves 80.7% mAP on COCO 2014 and 44.0% mAP on O365. Compared to training on human-annotated data, the performance gap is 2.6% on COCO and 4.6% on O365. Notably,

---

[1] The 80 categories in COCO 2014 were selected by the co-authors through voting based on how commonly the categories appear (Lin et al., 2014).

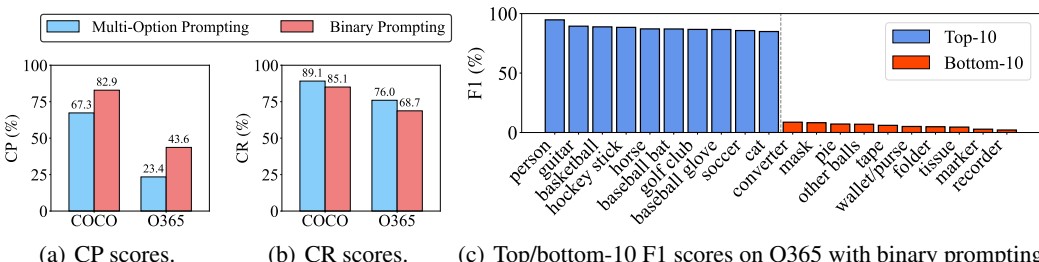

(a) CP scores.  (b) CR scores.  (c) Top/bottom-10 F1 scores on O365 with binary prompting.

Figure 2: The quality of annotations generated by Qwen2.5-VL.

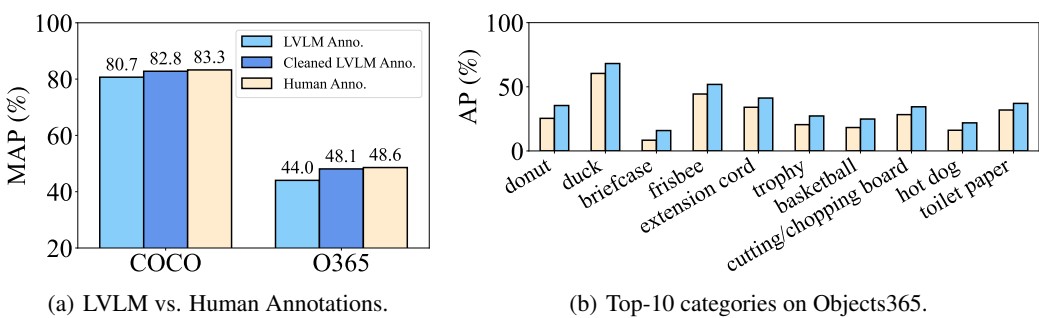

(a) LVLM vs. Human Annotations.  (b) Top-10 categories on Objects365.

Figure 3: The performance of ResNet-101 trained on LVLM-generated (by Qwen2.5-VL 7B) and human annotations.

the performance gap on O365 is nearly twice as large as that on COCO 2014, indicating greater challenges in handling uncommon/ambiguous categories when using LVLMs for annotation. Overall, the current level of annotation quality cannot yet be considered comparable to human performance, and there remains significant room for improvement.

**Finding 2: LVLM-generated annotations exhibit broad coverage.** Even when some labels are missing, the remained true labels are often sufficient to train a model whose performance is close to that trained with human-annotated data. This observation is further validated by Figure 2(b) and Figure 3(a), where we compare the performance of models trained on cleaned LVLM annotations (by filtering false positive labels according to human annotations) with those trained on human-annotated data [2]. From Figure 2(b), LVLM-generated annotations cover 89.1% and 76.0% of the human-annotated labels on COCO 2014 and O365, respectively. Interestingly, in Figure 3(a), models trained on these cleaned LVLM-generated labels achieve performance that is only 0.5% lower than those trained on human-annotated labels. These results suggest that, although LVLM-generated annotations can be incomplete, their broad coverage is sufficient to support effective model training.

**Finding 3: For certain categories, models trained on LVLM-annotated data surprisingly outperform those trained on human-annotated data.** Figure 3(b) illustrates the top-10 categories on O365, ranked by the performance difference between models trained on LVLM-annotated data and those trained on human-annotated data. We observe that for many categories, models trained on LVLM-annotated data significantly outperform those trained on human-annotated data. Unlike human annotators, whose performance may suffer from inattention or fatigue, LVLMs provide consistent annotations free from such human-induced variability. This further highlights the potential of LVLMs as a scalable and reliable alternative to human annotators.

## 3 THE LVLMANT FRAMEWORK

In this section, we introduce our LVLMANT framework, which is designed to generate human-comparable annotations. As shown in Figure 5, the framework mainly consisting of three components: (i) Prompts to candidates method , which uses the group-wise prompting and ensem-

---

[2]In this paper, "human-annotated" refers to the original annotations provided in the datasets. Note that these annotations were precisely-annotated by human annotators (see Appendix A for detailed information).

Figure 4: Comparison of three prompting strategies on COCO and Objects365. Left: Precision; middle: Recall; right: average number of candidate labels per image. DP (Disco-occurrence Partition) prompting yields the highest recall.

ble techniques to generate candidate label sets; (ii) LVLM-driven disambiguation, which develops the concept-aligned disambiguation strategy to identify true positives from the candidate sets; (iii) Human-assisted calibration, which is an optional choice to refine the candidate label set based on human feedback.

### 3.1 FROM PROMPTS TO CANDIDATES: GROUP-WISE PROMPTING AND LVLM ENSEMBLE

As discussed in Section 2.2, *binary* prompting typically yields high precision and recall. However, they require $q$ inferences per image, which is substantially more than single inference needed by *multi-option* prompting. By rough estimation, *binary* prompting takes at least ten times longer than *multi-option* prompting. For example, on COCO 2014 with Qwen2.5-VL 7B, *binary* prompting took over 153 hours, whereas *multi-option* prompting required only 10 hours. This resource gap increases with the number of categories and often makes *binary* prompting impractical when the label space comprises hundreds or thousands of categories. This observation motivates a two-stage strategy: first apply *multi-option* prompts to obtain a candidate set, and then apply *binary* prompts to disambiguate and prune false positives. We refer to the first step as P2C (Prompts to Candidates). *Multi-option* prompting generates a broad candidate set at low cost, thereby reducing the search space. *Binary* prompting then confirms category presence with high precision, pruning spurious labels. The combination attains high recall and high precision with improved cost-effectiveness.

**Divide-and-Conquer Prompting** When the number of classes is large, *multi-option* prompting faces an additional challenge: the prompt containing all candidate classes becomes long, which increases the risk of hallucinations and omissions. To address this, we adopt a divide-and-conquer strategy: partition the label space into multiple groups, perform inference for each group separately, and then merge the group-wise annotations. Specifically, we provide all categories to ChatGPT and prompt it to partition them into co-occurring and disco-occurring groups based on their category co-occurrence relationships. We consider two partitioning strategies: *Co-occurrence* Partition (CP), which groups classes that frequently appear together, and *Disco-occurrence* Partition (DP), which groups categories that rarely appear together. Figure 4 presents the results of three prompting strategies using Qwen2.5-VL on COCO 2014 and O365 (similar results on InternVL3 can be found in Appendix B.2). Surprisingly, DP prompting achieves higher recall than CP prompting. This finding appears counterintuitive, given that the auto-regressive nature of LVLMs should, in principle, allow CP to exploit within-group co-occurrence relationships, thereby improving recall. One plausible explanation is that grouping co-occurring categories increases the likelihood that multiple ground-truth labels appear in the same group. Within-group competition then suppresses weaker yet correct categories, which are consequently mislabeled. In contrast, disco-occurrence groups often contain only a single true label, so within-group competition is negligible.

**LVLM-Ensemble Annotation** To further ensure that more true labels are included in the candidate set, we propose an LVLM-ensemble strategy to generate candidate labels. Specifically, for each image $\boldsymbol{x}_i$, we input it into the $j$-th LVLM $\mathcal{L}_j$ for $j \in [m]$, where $m$ denotes the total number of LVLMs used, and obtain the corresponding responses as $R_i^j = \mathcal{L}_j(\boldsymbol{x}_i) = \{r_{ik}^j\}_{k=1}^l$, where $l$ is the number of disco-occurrence groups. Note that each response $r_{ik}^j$ is a sentence listing the annotated categories, separated by a delimiter (*e.g.*, a comma). Next, the task is to aggregate the responses from multiple LVLMs to construct a candidate label set for each image. Given that the primary goal in this stage is to ensure that annotation recall is sufficiently high, we adopt a straightforward approach to take the union over all responses.

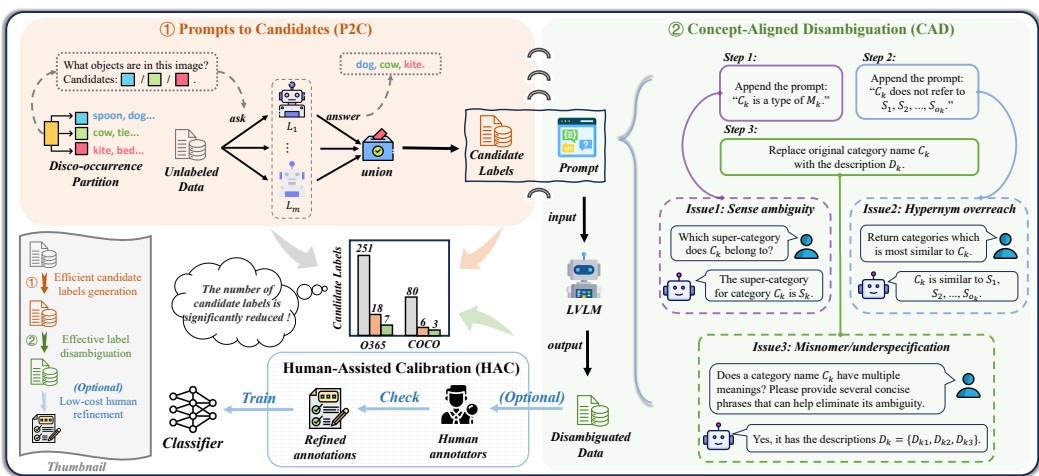

Figure 5: Illustration of the LVLMANT framework and HAC. The framework consists of two components: P2C and CAD. The former efficiently produces a high-recall candidate set, while the latter effectively eliminates false positives within this set to deliver high-precision annotations..

## 3.2 CONCEPT-ALIGNED DISAMBIGUATION

Candidate label disambiguation can be approached from three main directions: (i) LVLM-driven methods, which refine candidate labels through prompt design or LVLMs fine-tuning; (ii) Model-based methods, which train models on candidate labels and rely on their generalization capabilities to identify false positive labels; and (iii) human-assisted methods, which incorporate human feedback to refine candidate labels. Most existing PML studies focus on model-based methods (Xie & Huang, 2018; Zhang & Fang, 2020; Yang et al., 2024). Although these methods benefit from utilizing the useful information hidden in the candidate sets, they are also prone to overfitting to false positives, which can degrade disambiguation performance. In the proposed framework, we focus on LVLM-driven and human-assisted disambiguation.

False positive labels in the candidate set are typically attributed to object hallucination (Li et al., 2023; Leng et al., 2024) (or category hallucination) in LVLMs, where the model incorrectly predicts the presence of objects that are not actually present in the image. Our analysis reveals that object hallucination is caused not only by the inherent recognition limitations of LVLMs, but also by several external factors. An important factor is the misalignment between category names and intended semantic concepts, especially for those uncommon or ambiguous classes. We summarize three types of category-concept misalignments as follow:

- **Sense ambiguity** refers to cases where a category name is semantically unclear and may carry multiple interpretations. For example, the category name *orange*, which typically refers to the fruit but can also denote the color. These semantic ambiguities frequently cause the model to assign a large number of false positives, ultimately reducing the overall precision.

- **Hypernym overreach** refers to cases where a category name covers an excessively wide range of meanings. For example, in the COCO 2014 and O365 datasets, the category name *tie* typically refers to a *necktie*. However, the word *tie* can also broadly denote any knot-like object, leading to unintended over-generalization in LVLM annotation.

- **Misnomer/underspecification** refers to the use of category names that fail to precisely capture the intended semantic meaning. For example, in O365, the category names *mask* and *marker* are used to denote *face mask* and *marker pen*, respectively.

These three types of category-concept misalignment often cause LVLMs to generate excessive false positives, thereby reducing annotation precision and adversely affecting the performance of models trained on LVLM-annotated data. To mitigate this issue, we propose a Concept-Aligned Disambiguation (CAD) strategy, which aims to leverage LVLMs to refine the candidate label set by de-

signing prompts that are unambiguous and semantically clear. Specifically, this method consists of three main steps:

- For each category $C_k$, we ask ChatGPT-4o to identify the super-category for each category by providing the following prompt: `Which super-category does` $C_k$ `belong to? For example, an apple is a type of fruit, and a car is a type of vehicle.` After obtaining the super-category for each category $C_k$, denoted as $M_k$, we incorporate the following sentence into the annotation prompt: $C_k$ `is a type of` $M_k$`.`.

- For each category $C_k$, we ask ChatGPT-4o to return up to five categories from the entire category set whose visual appearance is most similar to $C_k$. We define the resulting set of similar categories as $\{S_j\}_{j=1}^{o_k}$, where $o_k$ denotes the number of similar categories. We then append the following sentence to the annotation prompt: $C_k$ `does not refer to` $S_1$`,` $S_2$`,` `...,` `or` $S_{o_k}$`.`.

- For each category $C_k$, we prompt ChatGPT-4o with the question: `Does a category name` $C_k$ `have multiple meanings? If so, please provide several concise phrases that can help eliminate its ambiguity.`.
  ChatGPT-4o will return a set of phrases based on both semantic relevance and common usage frequency. In real-world scenarios, users can flexibly choose suitable phrases to compose the description $D_k$ according to their preferences or requirements. For consistency in our experiments, we constructed $D_k$ using the top three phrases ranked by ChatGPT-4o. Then, we substitute the original category name $C_k$ in the annotation prompt with $D_k$, yielding the refined prompt: `Please carefully examine whether the image truly contains a` $D_k$`. Only answer No if you are very certain. Otherwise, answer Yes.`.

### 3.3 HUMAN-ASSISTED CALIBRATION

An alternative approach involves leveraging human feedback to refine the candidate label set. Unlike traditional manual annotation, which requires annotating the entire label set, our proposed Human-Assisted Calibration (HAC) only requires humans to verify a candidate label set that is orders of magnitude smaller. Specifically, using the P2C method proposed in Section 3.1, the average number of candidate labels that need to be checked per image is 5.66 for COCO 2014 and 18.16 for O365. Furthermore, if the candidate sets are further refined by the CAD method introduced in Section 3.2, the number of labels that require human verification is reduced to 2.96 for COCO 2014 and 6.86 for O365. In Section 4, we will present experimental results to demonstrate both model performance and the scale of cost reduction in human annotation.

## 4 EXPERIMENTS

In this section, we first present the main experimental results, followed by further analyses. Due to space limit, experimental settings are provided in the Appendix A.

### 4.1 COMPARISONS BETWEEN LVLMANT AND HUMAN ANNOTATORS

Table 1 and Table 2 report comparisons of LVLMANT with the state-of-the-art methods and human annotations in terms of both annotation quality and downstream model performance on COCO 2014 and O365. We can see that our proposed LVLMANT consistently outperforms the other methods in terms of both annotation quality and model performance. The resulting performance is only about 0.5% and 1.6% lower in mAP compared to using human annotations on COCO 2014 and O365, respectively. Incorporating human feedback via HAC further improves annotation quality and model performance. Compared to manual annotation, using the proposed methods, LVLMANT and P2C, significantly reduces annotation costs. On COCO 2014, the number of labels requiring annotation drops to 5.66 and 2.96, reducing the cost by approximately 14× and 27×. On O365, the numbers are 18.16 and 6.86, corresponding to 14× and 36× reductions. We also report the GPU time required for annotation with each method. Although the discriminative methods (RAM and RAM++) require the least time, they obtain unfavorable annotation performance. The proposed LVLMANT is markedly more efficient than the other methods.

Table 1: The results of annotation quality (OP, OR, OF1, CP, CR, CF1) and model performance (mAP) on COCO 2014. # C.L. denotes the average number of candidate labels per image that require manual annotation. # T. (h) denotes the GPU time in hours required by each method.

| | # C.L. | # T. (h) | OP | OR | OF1 | CP | CR | CF1 | mAP |
|---|---|---|---|---|---|---|---|---|---|
| Qwen2.5-VL 7B | – | 153 | 83.59 | 85.36 | 84.47 | 82.89 | 85.06 | 83.96 | 80.76 |
| Qwen2.5-VL 32B | – | 212 | 89.67 | 82.27 | 85.81 | 89.21 | 81.46 | 85.16 | 81.40 |
| RAM | – | < 1 | 89.24 | 55.27 | 68.26 | 89.92 | 62.14 | 73.49 | 79.48 |
| RAM++ | – | < 1 | 89.23 | 55.08 | 68.12 | 89.53 | 61.66 | 73.03 | 79.05 |
| NXTP | – | 4 | 62.78 | 55.94 | 59.16 | 57.89 | 50.43 | 53.91 | 57.38 |
| CLIP | – | < 1 | 59.30 | 59.20 | 59.25 | 65.00 | 61.65 | 63.28 | 66.85 |
| TagCLIP | – | 4 | 69.06 | 68.74 | 68.90 | 67.98 | 65.29 | 66.61 | 73.61 |
| CaSED | – | 3 | 86.22 | 23.85 | 37.36 | 83.30 | 28.70 | 42.69 | 54.42 |
| LVLMANT 7B | – | 15 | 85.74 | 86.37 | 86.06 | 84.65 | 86.52 | 85.57 | 82.69 |
| LVLMANT 32B | – | 66 | 85.65 | 86.81 | 86.23 | 85.09 | 86.35 | 85.71 | 82.74 |
| P2C + HAC | 5.66 | – | 100.00 | 96.79 | 98.37 | 100.00 | 97.00 | 98.46 | 83.34 |
| LVLMANT + HAC | 2.96 | – | 100.00 | 86.37 | 92.69 | 100.00 | 86.52 | 92.77 | 82.84 |
| Human Annotator | 80 | – | | – | | | – | | 83.26 |

Table 2: The results of annotation quality (OP, OR, OF1, CP, CR, CF1) and model performance (mAP) on O365. # C.L. denotes the average number of candidate labels per image that require manual annotation. # T. (h) denotes the GPU time in hours required by each method.

| | # C.L. | # T. (h) | OP | OR | OF1 | CP | CR | CF1 | mAP |
|---|---|---|---|---|---|---|---|---|---|
| Qwen2.5-VL 7B | – | 655 | 52.44 | 68.63 | 59.45 | 43.33 | 68.62 | 53.12 | 44.03 |
| Qwen2.5-VL 32B | – | 862 | 63.45 | 64.06 | 63.76 | 53.13 | 62.46 | 57.42 | 45.00 |
| RAM | – | < 1 | 66.15 | 23.70 | 34.90 | 55.95 | 31.68 | 40.45 | 40.09 |
| RAM++ | – | < 1 | 66.11 | 23.54 | 34.72 | 54.71 | 31.25 | 39.78 | 39.43 |
| NXTP | – | 4 | 34.71 | 25.11 | 29.14 | 41.72 | 18.52 | 25.65 | 23.81 |
| CLIP | – | < 1 | 39.60 | 40.87 | 40.22 | 31.90 | 29.94 | 30.89 | 27.75 |
| TagCLIP | – | 9 | 51.71 | 50.53 | 51.11 | 38.01 | 31.99 | 34.74 | 33.34 |
| CaSED | – | 3 | 63.84 | 6.41 | 11.65 | 48.02 | 11.12 | 18.05 | 22.08 |
| LVLMANT 7B | – | 67 | 60.43 | 68.52 | 64.22 | 46.99 | 67.79 | 55.50 | 46.47 |
| LVLMANT 32B | – | 246 | 63.71 | 68.92 | 66.21 | 49.52 | 66.72 | 56.85 | 46.99 |
| P2C + HAC | 18.16 | – | 100.00 | 84.87 | 91.81 | 100.00 | 87.06 | 92.61 | 48.89 |
| LVLMANT + HAC | 6.86 | – | 100.00 | 68.52 | 81.32 | 100.00 | 67.79 | 80.80 | 47.85 |
| Human Annotator | 251 | – | | – | | | – | | 48.58 |

To assess the practical effectiveness of LVLMANT, we conduct experiments on the COCO 2017 unlabeled image set and report performance on the COCO 2014 validation set. Figure 6 shows the overall mAP and the per class AP for all methods. Due to space limit, we display only the 20 categories where LVLMANT achieves the largest gains over the baselines, in order to analyze the sources of its advantages. We can see that LVLMANT achieves the best performance and significantly outperforms the comparing methods. LVLMANT substantially improves disambiguation for ambiguous categories, such as *orange*, *tie*, and *apple*, yielding high-quality annotations.

## 4.2 FURTHER STUDIES

To understand why the proposed method achieves high-quality annotations, we conduct ablation studies on the three disambiguation steps introduced in Section 3.2. Table 3 presents the results of these experiments on the O365 dataset. As shown in the table, each of the three disambiguation strategies effectively identifies false positive labels from the candidate set. Importantly, they significantly improve annotation precision with minimal cost in recall (*i.e.*, a slight drop in the OR and CR metrics), and ultimately contribute to better model training performance.

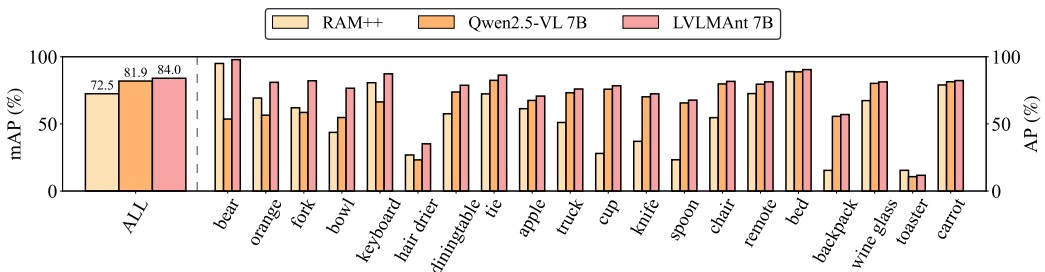

Figure 6: The performance on COCO 2014 validation set of ResNet-101 trained on annotations generated by difference methods on COCO 2017 unlabeled set.

Table 3: Abalation Studies on O365. S.-C. denotes super-category.

| P2C | S.-C. | Similarity | Description | OP | OR | OF1 | CP | CR | CF1 | mAP |
|-----|-------|------------|-------------|-------|-------|-------|-------|-------|-------|-------|
| ✓ | | | | 28.27 | 84.87 | 42.41 | 18.54 | 87.06 | 27.66 | 39.63 |
| ✓ | ✓ | | | 56.22 | 72.12 | 63.19 | 42.24 | 71.68 | 53.15 | 46.08 |
| ✓ | ✓ | ✓ | | 56.72 | 70.82 | 62.99 | 43.15 | 71.12 | 53.71 | 46.29 |
| ✓ | ✓ | ✓ | ✓ | 60.43 | 68.52 | 64.22 | 46.99 | 67.79 | 55.50 | 46.99 |

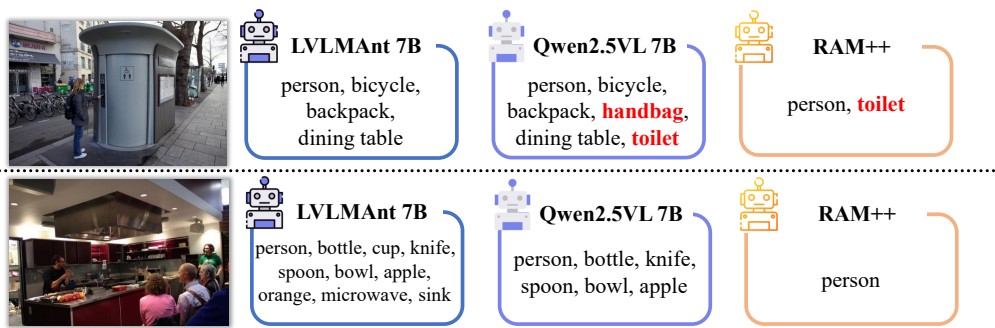

Figure 7: Visualization of annotations generated by diefferent methods on COCO 2017.

Finally, we present qualitative annotations on COCO 2017 produced by different methods. From Figure 7, RAM++ yields precise annotations but often misses many labels. Compared with Qwen2.5-VL, LVLMAnt achieves higher precision and broader coverage. These gains arise from the proposed P2C and CAD strategies: P2C improves efficiency and maximizes candidate coverage, while CAD performs effective disambiguation to improve precision.

## 5 RELATED WORKS

Recent studies have explored leveraging LLMs for data annotation. Wang et al. (2021) made the first attempt to leverage GPT-3 as a low-cost data annotator for natural language processing (NLP) tasks. (Ding et al., 2023) evaluated GPT-3 for annotating and augmenting data in classification and token-level tasks. Several studies have found that, on certain tasks, LLMs can surpass crowdsourced annotators in annotation quality (Gilardi et al., 2023; He et al., 2024). Wang et al. (2024) developed an annotation framework to leverage the strengths of both LLMs and humans to ensure the accuracy and reliability of annotations. Kim et al. (2024) developed MEGAnno+ system to facilitate human-LLM collaboration through efficient LLM annotation and selective human verification. Despite the great successes that these methods achieved, they mainly focused on NLP tasks and cannot be directly applied to image tagging tasks. Recognize Anything Model (RAM) (Zhang et al., 2024) and its improved version RAM++ (Huang et al., 2023) were proposed for image tagging tasks. However, their annotation quality still lags behind that of human annotators and therefore cannot replace human annotation.

## 6    CONCLUSION

This paper proposes an automated image tagging framework, LVLMANT, which integrates the group-wise prompting method to generate diverse candidate labels and employs the concept alignment mechanism to resolve semantic ambiguities through iterative refinement. The group-wise prompting approach ensures broad coverage of potential labels, whereas the alignment strategy dynamically calibrates category semantics using advanced language models to mitigate label-concept mismatches. Extensive experiments on benchmark datasets demonstrate the effectiveness of LVLMANT in achieving high-quality annotations comparable to human efforts, with significant reductions in annotation costs. A limitation of this work is that we only evaluate LVLMANT on natural image datasets. In future work, we plan to extend our evaluation to domain-specific datasets, such as those in the fine-grained image tagging domain.

## ETHICS STATEMENT

This paper does not raise any ethical concerns.

## REPRODUCIBILITY STATEMENT

The code of LVLMANT is available in supplementary material. The details of experimental settings are presented in Appendix A.

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

Table 4: The statics of benchmark datasets used.

| Dataset | # CLS | # Train | # Val | # Labels / Image | |
| --- | --- | --- | --- | --- | --- |
| | | | | Min~Max | Avg |
| COCO-2014 | 80 | 82,081 | 40,137 | 1~18 | 2.9 |
| Objects365-filtered | 251 | 104,395 | 78,303 | 1~30 | 6.3 |
| COCO-2017-unlabeled | 80 | 123,403 | - | - | - |

## A  DETAILED EXPERIMENTAL SETTINGS

**Datasets**  To evaluate the performance of the proposed method, we perform experiments on three benchmark datasets, including MS-COCO 2014 (COCO 2014), MS-COCO 2017 (COCO 2017), and Objects365 (O365). We perform several pre-processing steps on all datasets. We utilize the original train/validation split from the 2014 version of the COCO [3] dataset, which contains 82,081 training images and 40,137 validation images for 80 classes, with an average of 2.9 labels per image. O365 [4] presents significant annotation challenges, due to its large scale, encompassing 365 categories and more than 1.8 million images. Annotating such a large number of images using large models is highly challenging and time-consuming. To improve efficiency, we randomly sampled a subset of approximately 100,000 images from the full dataset. In addition, we applied a preprocessing step to retain only the labels with more than 100 positive instances. This step is important for accurately assessing the impact of annotation quality. We found that if labels with very few positive samples are kept, their consistently poor performance, regardless of annotation quality, can reduce the apparent influence of annotation quality on overall model performance. In such cases, the presence of extremely rare labels makes it difficult to observe meaningful differences, since the model tends to perform poorly on these labels no matter how well they are annotated. This results in 104,395 training images and 78,303 validation images, covering 251 categories, with an average of 6.3 labels per image. For COCO 2017 in particular, we use the unlabeled split as a practical application scenario for our method, which contains 123,403 images without manual annotations. Since ground-truth labels are unavailable, we cannot directly score annotation quality. Instead, we train models on the annotations produced by each method and assess annotation quality via their performance on the COCO 2014 validation set. The detail characteristics of three benchmark datasets are reported in Table 4.

**Evaluation Metrics**  We employs two types of evaluation metrics to assess annotation quality and model performance, respectively. For annotation quality, we use six metrics: Overall Precision (OP), Overall Recall (OR), Overall F1-score (OF1), per-Class Precision (CP), per-Class Recall (CR), and per-Class F1-score (CF1). Specifically, these metrics can be computed as follow:

$$\text{CF1} = \frac{2 \times \text{CP} \times \text{CR}}{\text{CP} + \text{CR}}, \quad \text{OF1} = \frac{2 \times \text{OP} \times \text{OR}}{\text{OP} + \text{OR}},$$

and

$$\text{CP} = \frac{1}{K} \sum_k \frac{N_k^{TP}}{N_k^{TP} + N_k^{FP}}, \quad \text{OP} = \frac{\sum_k N_k^{TP}}{\sum_k (N_k^{TP} + N_k^{FP})},$$

$$\text{CR} = \frac{1}{K} \sum_k \frac{N_k^{TP}}{N_k^{TP} + N_k^{FN}}, \quad \text{OR} = \frac{\sum_k N_k^{TP}}{\sum_k (N_k^{TP} + N_k^{FN})},$$

where CP, CR are average per-class precision, recall, and OP, OR are overall precision, recall. According to the confusion matrix, $\{N_k^{TP}, N_k^{FP}, N_k^{TN}, N_k^{FN}\}$ indicate the number of true positive, false positive, true negative, false negative for the $k$-th class.

To evaluate model performance, we use mean Average Precision (mAP) to measure the model trained under various annotation strategies. It is defined as follows:

$$\text{mAP} = \frac{1}{K} \sum_{k=1}^{K} \text{AP}_k, \quad \text{AP}_k = \text{AP}\left(\boldsymbol{p}_{\cdot k}, \boldsymbol{y}_{\cdot k}\right) = \frac{1}{|\boldsymbol{y}_{\cdot k}|} \sum_{i:y_{ik}=1} \frac{|\{y_{jk} = 1 : \boldsymbol{p}_{jk} > \boldsymbol{p}_{ik}\}|}{\textbf{rank}\left(\boldsymbol{p}_{\cdot k}\right)_i},$$

---

[3]https://cocodataset.org/#home
[4]https://www.objects365.org/overview.html

Table 5: Category difference comparison: Original dataset category to ($\rightarrow$) RAM category.

| COCO | | | Objects365 | | |
|---:|:---:|:---|---:|:---:|:---|
| airplane | $\rightarrow$ | plane | sneakers | $\rightarrow$ | running shoe |
| fire hydrant | $\rightarrow$ | hydrant | other shoes | $\rightarrow$ | shoe |
| suitcase | $\rightarrow$ | luggage | desk | $\rightarrow$ | table |
| skis | $\rightarrow$ | ski | street lights | $\rightarrow$ | street light |
| potted plant | $\rightarrow$ | plant | cabinet/shelf | $\rightarrow$ | cabinet |
| dining table | $\rightarrow$ | table | handbag/satchel | $\rightarrow$ | handbag |
| toilet | $\rightarrow$ | toilet bowl | picture/frame | $\rightarrow$ | picture |
| tv | $\rightarrow$ | television | gloves | $\rightarrow$ | glove |
| cell phone | $\rightarrow$ | smartphone | leather shoes | $\rightarrow$ | leather shoe |
| refrigerator | $\rightarrow$ | fridge | potted plant | $\rightarrow$ | plant |
| teddy bear | $\rightarrow$ | teddy | bowl/basin | $\rightarrow$ | bowl |
| | | | boots | $\rightarrow$ | boot |
| | | | monitor/tv | $\rightarrow$ | monitor |
| | | | trash bin can | $\rightarrow$ | can |
| | | | slippers | $\rightarrow$ | slipper |
| | | | barrel/bucket | $\rightarrow$ | barrel |
| | | | sandals | $\rightarrow$ | sandal |
| | | | pen/pencil | $\rightarrow$ | pencil |
| | | | wild bird | $\rightarrow$ | bird |
| | | | high heels | $\rightarrow$ | high heel |
| | | | cell phone | $\rightarrow$ | smartphone |
| | | | canned | $\rightarrow$ | drink |
| | | | lifesaver | $\rightarrow$ | life jacket |
| | | | awning | $\rightarrow$ | canopy |

where $\boldsymbol{p}$ and $\boldsymbol{y}$ denote the predicted probabilities and ground truth labels, respectively, and mAP is the mean of the average precision (AP) over all classes. The denominator $\mathbf{rank}\,(\boldsymbol{p}_{\cdot k})_i = 1 + |\{j : \boldsymbol{p}_{jk} > \boldsymbol{p}_{ik}\}|$ represents the rank of the model output $\boldsymbol{p}_{ik}$ among all predictions for class $k$, and the numerator $|\{y_{jk} = 1 : \boldsymbol{p}_{jk} > \boldsymbol{p}_{ik}\}|$ counts the number of true positive predictions that have a higher predicted probability than $\boldsymbol{p}_{ik}$.

**Comparing Methods**   To validate the effectiveness of LVLMANT, we compare it against the powerful LVLM, Qwen2.5-VL, the Recognize Anything Model (RAM) (Zhang et al., 2024) and its improved version RAM++ (Huang et al., 2023), NXTP (Yue et al., 2024), CLIP (Radford et al., 2021), TagCLIP (Lin et al., 2024) and vocabulary-free image classifier, CaSED (Conti et al., 2023). We also compare against human annotators (see Appendix A for detailed information). Below, we provide a detailed introduction to these compared methods.

- **Qwen2.5-VL** is a large-scale vision-language model developed by Alibaba Group, capable of supporting both multimodal understanding and generation tasks. It employs a dynamic-resolution vision encoder (ViT) and a cross-modal projector, integrated with a language model for end-to-end training.

- **NXTP** is a framework that formulates object recognition as a next-token prediction problem, inspired by the autoregressive modeling paradigm commonly used in large language models. Instead of treating object recognition as a multi-label classification task, NXTP views the set of objects in an image as a sequence of tokens and trains a model to predict the next object token conditioned on previously predicted ones.

- **CLIP** is a vision-language model proposed by OpenAI that learns to connect images and text through large-scale contrastive training. It is trained on hundreds of millions of image–text pairs collected from the web, enabling it to align visual representations with natural language descriptions.

- **TagCLIP** is an improved version of CLIP designed specifically for performing zero-shot multi-label image classification. Unlike standard CLIP, which focuses on matching an

image with a single text description, TagCLIP introduces a local-to-global Framework to exploit patch-level features, which provide more useful information for capturing multiple semantics.

- **CaSED** refers to the vocabulary-free image classification approach that does not rely on a predefined set of category names. Instead, it learns to classify images by identifying visual concepts directly from data, allowing the model to generalize beyond a fixed label vocabulary. This enables CaSED to handle open-vocabulary or emergent categories more effectively, especially in scenarios where the full set of possible classes is unknown or difficult to enumerate.

- **RAM** is a foundational model for image tagging that leverages large-scale automatically parsed tags from image-text pairs. Using a Swin Transformer as the image encoder and a lightweight tag recognition decoder, RAM achieves high-accuracy zero-shot recognition of over 6,400+ common categories. Without relying on manual annotations, the model constructs its training data through semantic text parsing and an automated tag-cleaning engine. It significantly outperforms general-purpose vision-language models like CLIP and BLIP on multiple image tagging benchmarks.

- **RAM++** is an enhanced version of RAM. Its core innovation lies in a multi-grained text supervision mechanism that integrates both global text supervision and individual tag supervision, further improving recognition of open-set categories. The model adopts a unified Image-Tag-Text Alignment (ITTA) framework and uses large language models (*e.g.*, GPT-3.5) to generate diverse descriptions for each tag, thereby expanding the model's capacity to recognize previously unseen categories and complex phrases (*e.g.*, human-object interactions). RAM++ achieves state-of-the-art zero-shot image tagging performance on datasets including OpenImages, ImageNet, and HICO.

These models represent different technical routes in multimodal annotation, covering a spectrum from general multimodal understanding (Qwen2.5-VL) to specialized image tag recognition (RAM/RAM++), providing comprehensive baselines for our experimental comparisons. To ensure compatibility with the label space of the RAM and RAM++ models, the categories from the COCO 2014 and O365 datasets must be appropriately mapped. The specific category correspondences with naming differences are summarized in Table 5. Categories not mentioned in the table remain unchanged, as their names are consistent with those in RAM's label space. As shown in the table, the mapping discrepancies can be categorized into three types: (1) singular vs. plural forms (*e.g.*, "gloves" → "glove"); (2) synonymous expressions (*e.g.*, "refrigerator" → "fridge"); and (3) near-synonyms expressions(*e.g.*, "desk" → "table"). This structured mapping ensures that the annotated labels are consistent with RAM's predefined categories, thereby facilitating smoother training and evaluation processes. RAM and RAM++ produce per-class probability scores. Consistent with the original papers, we obtain the final predictions by applying suggested thresholds. The class-specific thresholds are available in [5].

**Human Annotation**    The human annotations referred to in this paper are the original annotations provided in the respective datasets. Below, we briefly introduce the workflows and methodologies related to image category annotation in each dataset.

- **MS-COCO** Dataset: With 91 object categories (the 2014 release was limited to 80) and a large number of images, it would be prohibitively expensive to have workers answer all these binary classification questions per image. Therefore, the authors adopted a hierarchical annotation strategy. The object categories were grouped into 11 super-categories. Annotators were first asked to determine whether any instance from a given super-category (*e.g.*, "animal") was present in the image. If present, they then annotated specific subordinate categories (*e.g.*, "dog", "cat") within that super-category. To improve efficiency and accuracy, annotators placed category icons onto corresponding object instances in the image via a drag-and-drop interface. To ensure high recall of category labels, multiple annotators independently labeled each image, and the final determination of category presence was based on the union of all annotators' responses. Each image was independently

---

[5]`https://github.com/xinyu1205/recognize-anything/blob/main/ram/data/ram_tag_list_threshold.txt`.

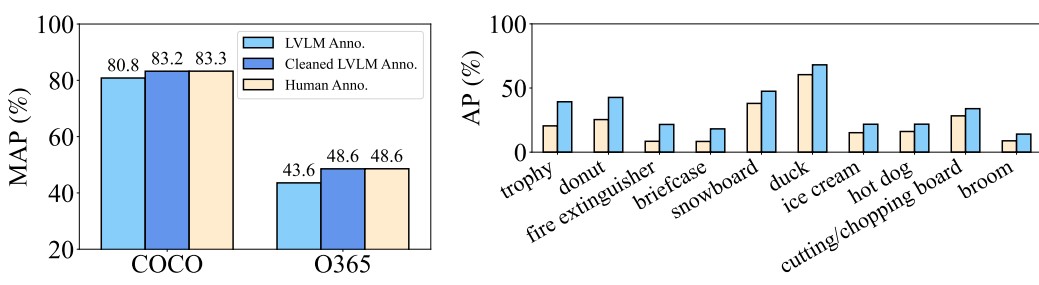

(a) LVLM vs. Human Annotations.

(b) Top-10 categories on Objects365.

Figure 8: The performance of ResNet-101 trained on LVLM-generated (by InternVL3 8B) and human annotations.

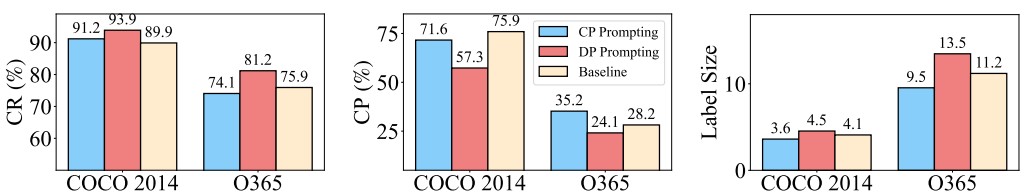

Figure 9: Comparison of three prompting strategies using InternVL3 8B on COCO and Objects365. Left: Recall; middle: Precision; right: average number of candidate labels per image. DP (Disco-occurrence Partition) prompting yields the highest recall.

annotated by 8 annotators via Amazon Mechanical Turk. To ensure annotation accuracy, false positives identified during this stage were handled in the subsequent instance segmentation stage, where workers could indicate that no instance of a given category was present in the image.

- **Objects365** Dataset: The annotation pipeline was designed to efficiently handle a large number of categories. The final dataset contains 365 object categories, selected based on frequency from an initial candidate set of 442 categories, and organized under 11 predefined super-categories (*e.g.*, "clothes", "kitchen"). The annotation process began with a binary classification step to filter out iconic images or those containing none of the target objects. Images that passed this filter proceeded to the image-level tagging stage, where annotators identified one or more super-categories present in the image. This phased and super-category-driven approach significantly reduced the cognitive load on individual annotators, who only needed to be familiar with approximately 30-40 categories within a specific super-category, thereby ensuring feasibility and consistency in large-scale category annotation. The entire process was carried out by a professional team consisting of Annotators (who were required to complete a training course and pass an examination before starting annotation), Inspectors, and Examiners, working in coordination to ensure high-quality annotation results.

Overall, both COCO and Objects365 employed similar hierarchical annotation strategies, grouping fine-grained categories into super-categories and performing annotation in a staged manner. Furthermore, both datasets implemented validation mechanisms to ensure annotation accuracy.

## B  ADDITIONAL EXPERIMENTAL RESULTS

### B.1  COMPARISON BETWEEN LVLM AND HUMAN

Figure 8(a) illustrates performance of models trained on LVLM, clean LVLM (by filtering false positives according to human annotations), and human annotations. Figure 8(b) illustrates the top-10 categories on O365, ranked by the performance difference between models trained on LVLM-annotated data and those trained on human-annotated data. We use InternVL3 8B for this experiment. From

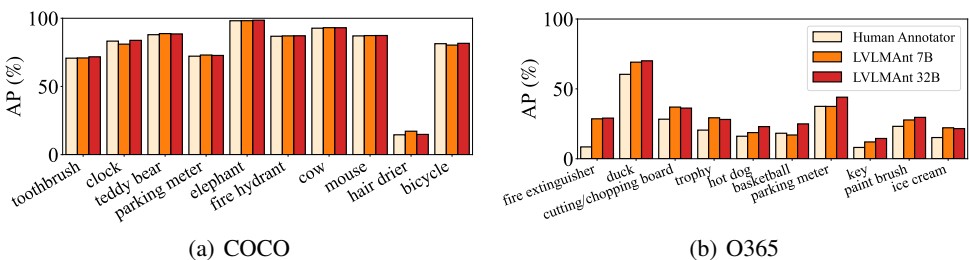

(a) COCO  (b) O365

Figure 10: Performance of models trained on LVLMANT-generated annotations and human annotations.

the figure, we obtain conclusions similar to those of the corresponding experiment reported in the main text (using Qwen2.5-VL 7B).

### B.2 COMPARISON OF PROMPTING METHODS

Figure 9 presents the results of three prompting strategies using InternVL3 on COCO 2014 and O365. Consistent with the Qwen2.5-VL results, DP prompting achieves significantly higher recall than the other two methods, but exhibits lower precision.

### B.3 FURTHER STUDIES

To demonstrate the potential of using LVLMANT as a substitute for human annotators, Figure 10 illustrates the classes for which training on LVLMANT-annotated data achieves the largest improvements in performance over human-annotated data, specifically the top 10 classes on COCO and O365. As shown in the figure, for the COCO dataset, which primarily consists of common categories, LVLMANT shows advantages only in a few challenging classes. For example, the class *hair drier* contains very few positive instances, making it difficult for human annotators. For the O365 dataset, which includes a large number of uncommon or ambiguous categories, LVLMANT demonstrates significant advantages over human annotations across multiple classes. These results demonstrate the potential of LVLMANT to achieve human-level ability in image tagging.

## C DISCUSSION ON P2C

After using P2C (introduced in Section 3.1) to obtain candidate label sets, the problem is related to a weakly supervised learning scenario called PML (Xie & Huang, 2018; Zhang & Fang, 2020). In PML, each instance is associated with a candidate label set, which contains all ground-truth labels and other noisy (false positive) labels. One subtle difference is that PML typically assumes that the candidate label set contains all relevant labels, whereas the LVLM-generated candidate sets often fail to achieve complete coverage of all relevant labels. However, we find that this incomplete coverage is not critical, as models trained on the cleaned LVLM-generated labels can achieve performance comparable to, or even surpassing, that of models trained on human-annotated data. One of our contributions is that the proposed LVLM-ensemble candidates generation method enables the construction of realistic PML datasets with practical significance, unlike the previous works that often conducted experiments on synthetic PML data.

