# OpenReview forum: "Are Large Vision-Language Models Good Annotators for Image Tagging?"
_ICLR.cc/2026/Conference — Submitted to ICLR 2026_

### Official Review · Reviewer_C8ze · 2025-10-28

**Soundness:** 3
**Presentation:** 3
**Contribution:** 2
**Rating:** 4
**Confidence:** 4

**Summary:**

This paper investigates if LVLMs can be used as an alternative to Human annotations for image tagging task. The paper presents a systematic analysis comparing LVLM-generated annotations with human annotations on MS-COCO 2014 and Objects365 datasets. They find that  LVLMs annotations are: (1) accurate for common categories (and perform better than human annotations) but fail with uncommon categories. (2) Robust to missing labels (even if LVLMs miss labels, the other labels are often sufficient to train the model). Based on these, the paper presents LVLMANT, a two-stage framework consisting of: (1) Prompts-to-Candidates (P2C) using group-wise prompting and ensemble techniques to generate candidate labels, and (2) Concept-Aligned Disambiguation (CAD) to refine candidates by addressing semantic misalignments.

**Strengths:**

1. Paper provides comprehensive empirical analysis of different prompting strategies (open-ended, multi-option, binary) and their trade-offs.

2. LVLMANT effectively combines the efficiency of multi-option prompting with the precision of binary prompting, achieving a good balance between annotation quality and cost.

3. Extensive experiments on multiple benchmark datasets with detailed ablations is provided

4. The observation that models trained on LVLM annotations sometimes outperform those trained on human annotations for certain categories provides valuable perspective

**Weaknesses:**

1. How are the sets of co-occurring and disco-occurring classes chosen?

2. Following up on the previous point, more discussion on cases where LVLMANT fails needs to be discussed. For example, If co-occurrence patterns are derived from ChatGPT4, do these align with actual patterns in datasets like COCO? The paper lacks systematic analysis of whether pattern misalignment degrades performance.

3. Testing with weaker LVLMs (like LLava or InstructBLIP) would better demonstrate robustness, particularly whether CAD can handle noisier candidate sets from lower-capability models.

4. I like the analysis provided by the paper, but I believe the work is more applied and could be a better fit at vision conferences (CVPR, ICCV/ECCV) or NLP conferences. Based on "call for papers" I could not find a good match for it.

**Questions:**

Please see the sections above.

---

> ### Author Response · Authors · 2025-11-21
>
> Thanks for your constructive comments. We are glad that you considered our work “comprehensive empirical analysis, good balance between annotation quality and cost”. We are glad to answer all your questions.
>
> **Q1**: How are the sets of co-occurring and disco-occurring classes chosen?
>
> **R1**:  We provide all categories to ChatGPT and prompt it to **partition them into co-occurring and disco-occurring groups** based on their **category co-occurrence relationships**. We have added this clarification in the rebuttal version.
>
> **Q2**: Following up on the previous point, more discussion on cases where LVLMANT fails needs to be discussed. For example, If co-occurrence patterns are derived from ChatGPT4, do these align with actual patterns in datasets like COCO? The paper lacks systematic analysis of whether pattern misalignment degrades performance.
>
> **R2**: Thank you for the suggestion.  *Figure 4* can be regarded as a partial discussion on **how the degree of intra-group co-occurrence affects annotation performance**. The disco-occurrence partition (DP) generated by ChatGPT can be viewed as **nearly disco-occurring**, since most categories rarely co-occur, making it straightforward to obtain such a partition. In contrast, the co-occurrence partition (CP) produced by ChatGPT can be considered an approximation of the true partition, which is derived from true labels. Although it is not perfectly aligned with the true partition, the intra-group co-occurrence in CP is substantially higher than in DP.
>
> From the results in *Figure 4*, we can observe that **the degree of intra-group co-occurrence has a strong impact on annotation performance**. The DP method shows **high recall but low precision**, while the CP method shows the opposite trend, exhibiting **low recall but high precision**. This indicates that **a higher level of intra-group co-occurrence tends to reduce the recall** of model-based annotation.
>
> Since the primary goal of the P2C stage is to achieve high recall, we adopt the DP strategy for annotation in the LVLMAnt framework. This indicates that whether the CP generated by ChatGPT perfectly align with the true dataset partitions **does not affect the annotation performance**.
>
> **Q3**: Testing with weaker LVLMs (like LLava or InstructBLIP) would better demonstrate robustness, particularly whether CAD can handle noisier candidate sets from lower-capability models.
>
> **R3**:  Thank you for the suggestion. Table 7 reports the results obtained using LLaVA-Next, which is a significantly weaker LVLM compared with Qwen2.5-VL. Since LLaVA-Next does not support batch inference, using it for annotation is time-consuming. Due to time limit, we only conducted experiments on COCO 2014. As shown in the table, even when using a **relatively weaker LVLM**, our proposed method still **significantly improves annotation quality**, which in turn **enhances the performance of downstream training models**. We will further improve and extend this part of the experiments in the future version.
>
> Table 7. The results of annotation quality and model performance on COCO 2014.
>
> | COCO 2014               | OP    | OR    | OF1   | CP    | CR    | CF1   | mAP   |
> | ----------------------- | ----- | ----- | ----- | ----- | ----- | ----- | ----- |
> | LLaVA-Next 8B           | 68.94 | 89.85 | 78.02 | 69.07 | 89.89 | 78.12 | 79.04 |
> | RAM                     | 89.24 | 55.27 | 68.26 | 89.92 | 62.14 | 73.49 | 79.48 |
> | RAM++                   | 89.23 | 55.08 | 68.12 | 89.53 | 61.66 | 73.03 | 79.05 |
> | LVLMAnt (LLaVA-Next 8B) | 77.08 | 87.93 | 82.15 | 74.71 | 87.84 | 80.74 | 80.95 |

---

> > ### Author Response · Authors · 2025-11-21
> >
> > **Q4**: I like the analysis provided by the paper, but I believe the work is more applied and could be a better fit at vision conferences (CVPR, ICCV/ECCV) or NLP conferences. Based on "call for papers" I could not find a good match for it.
> >
> > **R4**: We appreciate the reviewer’s positive feedback and the suggestion regarding the venue fit. We would like to emphasize that the primary contribution of our paper lies in **addressing one of the most fundamental challenges in traditional machine learning: the annotation bottleneck**, where every new task requires a new set of labeled data. This challenge is central to the vision of **data-centric AI**.
> >
> > Our work targets one of the most common and foundational multimodal tasks, **image tagging**, also known as **multi-label classification**, and aims to alleviate the annotation bottleneck in this setting. To this end, we first summarize three prompting strategies for LVLM-based annotation and provide a systematic analysis of the annotation performance of state-of-the-art open-source LVLMs. Our analysis reveals that these models perform well on common categories but struggle with uncommon or ambiguous ones, and even human annotators exhibit errors. This observation motivates us to develop an annotation framework that reduces manual annotation cost while improving annotation quality.
> >
> > We therefore propose LVLMAnt, a structured LVLM-based annotation framework consisting of two complementary stages:
> >
> > 1. a multi-option prompting stage that efficiently produces a compact candidate set and reduces annotation time, and
> > 2. a binary prompting verification stage that carefully refines these candidates and significantly enhances annotation quality.
> >    These two stages correspond to our core technical components, Prompt-to-Candidate (P2C) and Concept-Aligned Disambiguation (CAD).
> >
> > Compared with existing LVLM-based annotation methods, LVLMAnt achieves **both high efficiency and high annotation quality**, offering a **data-centric perspective** on improving LVLM-driven annotation. We believe this work aligns well with the conference’s emphasis on **data-centric AI**.

---

> > > ### Author Response · Authors · 2025-11-28
> > >
> > > Dear C8ze,
> > >
> > > Thank you again for taking the time to provide such insightful and constructive comments. We hope that our rebuttal addressed the main concerns you raised. If any of our explanations require further detail or if additional points should be discussed, we would be glad to elaborate during the ongoing discussion phase.
> > >
> > > Best,
> > > The Authors

---

### Official Review · Reviewer_HCdA · 2025-11-01

**Soundness:** 3
**Presentation:** 3
**Contribution:** 3
**Rating:** 4
**Confidence:** 4

**Summary:**

This paper addresses the problem of automating annotation for image tagging. To avoid notable labor and costs introduced by traditional human annotation, the work proposes to leverage large vision-language models for generating labels. They present two strategies, including Prompts-to-candidates (P2C) and Concept-Aligned Disambiguation (CAD), building the LVLMANT framework. P2C employes group-wise prompting and annotation ensembling to produce candicate sets, and CAD further calibrates the concept spaces. Extensive experiments on COCO and Object365 show the effectiveness of the proposed framework.

**Strengths:**

1. The writing and figure are clear, which makes the paper easy to follow and understanding.
2. Convincing analyses are presented. Metrics, including CP, CR, and downstream performance using the generated labels, verify the claims and show the efficacy of proposed P2C and CAD strategies.

**Weaknesses:**

1. Comparison with some other works are missing, like [1], CaSED [2], and CLIP. Like RAM and RAM+, they can also be used for annotation.
2. Some experimental details are not presented. In Table 1, what is the category vocabulary used by Qwen2.5-VL-7B/32B, RAM/RAM++, and LVLMANT 7B/32B? Is it the ground truth category set of datasets? Why the GPU time in hours required by Qwen2.5-VL are so large? Does Qwen2.5-VL use binary prompting? Besides, in the appendix, it is stated that categories are mapped for RAM++ and seems that predefined categories are used for RAM++. RAM++ also supports customizing tag categories for recognition. The baseline that directly using the categories from the datasets for RAM++ is missing.
3. In Table 3, the performance improvements of Similarity and Description are relatively small, whose effectiveness is not sufficiently verified.
4. The GPU time in hours for P2C and CAD are not presented, respectively, which makes the time saving inclear.
5. In Table 3, there lacks a baseline that P2C + binary promping, which makes the efficay of CAD unclear.

[1] Object Recognition as Next Token Prediction. CVPR 2024.

[2]  Vocabulary Free Image Classification. NeurIPS, 2023.

**Questions:**

Please see the Weaknesses

---

> ### Author Response · Authors · 2025-11-21
>
> Thanks for your constructive comments. We are glad that you considered our work “clear, easy to follow and understanding, convincing analyses”. We are glad to answer all your questions.
>
> **Q1**: Comparison with some other works are missing, like [1], CaSED [2], and CLIP. Like RAM and RAM+, they can also be used for annotation.
>
> **R1**: Thank you for the suggestion. In the tables below, we compare our method with the mentioned approaches, including NXTP [1] CLIP, TagCLIP [3] and CaSED [2]. TagCLIP is an improved version of CLIP designed for the multi-label scenarios. It can be seen that these methods show **a substantial gap** from ours in terms of both annotation quality and downstream model performance. It is worth noting that NXTP and CaSED do not support customized categories. As a result, most of its predicted categories may not exist within the dataset’s label space, leading to a relatively low recall. These results have been included in the rebuttal version. Please see Tables 1 and 2 in the rebuttal version for details.
>
> | COCO2014   | OP    | OR    | OF1   | CP    | CR    | CF1   | mAP   |
> | ---------- | ----- | ----- | ----- | ----- | ----- | ----- | ----- |
> | NXTP       | 62.78 | 55.94 | 59.16 | 57.89 | 50.43 | 53.91 | 57.38 |
> | CLIP       | 59.30 | 59.20 | 59.25 | 65.00 | 61.65 | 63.28 | 66.85 |
> | TagCLIP    | 69.06 | 68.74 | 68.90 | 67.98 | 65.29 | 66.61 | 73.61 |
> | CaSED      | 86.22 | 23.85 | 37.36 | 83.30 | 28.70 | 42.69 | 54.42 |
> | LVLMAnt 7B | 85.74 | 86.37 | 86.06 | 84.65 | 86.52 | 85.57 | 82.69 |
>
> | Objects365 | OP    | OR    | OF1   | CP    | CR    | CF1   | mAP   |
> | ---------- | ----- | ----- | ----- | ----- | ----- | ----- | ----- |
> | NXTP       | 34.71 | 25.11 | 29.14 | 41.72 | 18.52 | 25.65 | 23.81 |
> | CLIP       | 39.60 | 40.87 | 40.22 | 31.90 | 29.94 | 30.89 | 27.75 |
> | TagCLIP    | 51.71 | 50.53 | 51.11 | 38.01 | 31.99 | 34.74 | 33.34 |
> | CaSED      | 63.84 | 6.41  | 11.65 | 48.02 | 11.12 | 18.05 | 22.08 |
> | LVLMAnt 7B | 60.43 | 68.52 | 64.22 | 46.99 | 67.79 | 55.50 | 46.47 |
>
> [1] Object Recognition as Next Token Prediction.
>
> [2] Vocabulary Free Image Classification.
>
> [3] TagCLIP: A Local-to-Global Framework to Enhance Open-Vocabulary Multi-Label Classification of CLIP Without Training.
>
> **Q2**: In Table 1, what is the category vocabulary used by Qwen2.5-VL-7B/32B, RAM/RAM++, and LVLMANT 7B/32B? Is it the ground truth category set of datasets? Why the GPU time in hours required by Qwen2.5-VL are so large? Does Qwen2.5-VL use binary prompting?
>
> **R2**: For all methods, we use the category names provided in the dataset metadata. Qwen2.5-VL adopts binary prompting, which requires q inferences per image, resulting in a longer annotation time.
>
> **Q3**: In the appendix, it is stated that categories are mapped for RAM++ and seems that predefined categories are used for RAM++. RAM++ also supports customizing tag categories for recognition. The baseline that directly using the categories from the datasets for RAM++ is missing.
>
> **R3**: Thank you for the suggestion. Table 6 presents the experimental results of the RAM series methods **under the settings of  category mapping or using customized categories (denoted by CC)**. As shown in the table, both RAM and RAM++ perform better when category mapping is applied. These results are consistent with our expectations, as open-vocabulary object recognition is generally more challenging than closed-set object recognition. On the other hand, this also indicates that the category mapping is well designed.
>
> Table 6.  The results of the RAM series methods with and without customized categories on COCO2014 and Objects365. CC denotes the method use customized categories.
>
> | COCO2014   |  OP   |  OR   |  OF1  |  CP   |  CR   |  CF1  |  mAP  |
> | ---------- | :---: | :---: | :---: | :---: | :---: | :---: | :---: |
> | RAM        | 89.24 | 55.27 | 68.26 | 89.92 | 62.14 | 73.49 | 79.48 |
> | RAM++      | 89.23 | 55.08 | 68.12 | 89.53 | 61.66 | 73.03 | 79.05 |
> | RAM (CC)   | 94.80 | 45.45 | 61.44 | 92.85 | 54.94 | 69.03 | 69.84 |
> | RAM++ (CC) | 90.47 | 21.35 | 34.55 | 87.75 | 33.97 | 48.98 | 63.56 |
>
> | Objects365 |  OP   |  OR   |  OF1  |  CP   |  CR   |  CF1  |  mAP  |
> | ---------- | :---: | :---: | :---: | :---: | :---: | :---: | :---: |
> | RAM        | 66.15 | 23.70 | 34.90 | 55.95 | 31.68 | 40.45 | 40.09 |
> | RAM++      | 66.11 | 23.54 | 34.72 | 54.71 | 31.25 | 39.78 | 39.43 |
> | RAM (CC)   | 67.01 | 18.90 | 29.49 | 59.29 | 28.62 | 38.61 | 32.13 |
> | RAM++ (CC) | 36.49 | 8.15  | 13.33 | 46.03 | 23.71 | 31.30 | 28.83 |

---

> > ### Author Response · Authors · 2025-11-21
> >
> > **Q4**: In Table 3, the performance improvements of Similarity and Description are relatively small, whose effectiveness is not sufficiently verified.
> >
> > **R4**: This can be explained from two perspectives.
> >
> > 1. **CAD should be viewed as an integrated module**, where the minor contributions of its individual components collectively lead to a **substantial overall improvement**.
> > 2. The effectiveness of the LVLMAnt annotation framework also benefits from the **complementary nature** of **multi-option prompting** and **binary prompting**. These two annotation modes exhibit **different annotation preferences**, enabling them to **identify different types of irrelevant labels** and thereby achieve **better overall annotation performance**. We discuss this point in detail in **R6**, and in future work, we plan to conduct a more in-depth analysis to further investigate this phenomenon.
> >
> > **Q5**: The GPU time in hours for P2C and CAD are not presented, respectively, which makes the time saving inclear.
> >
> > **R5**: The goal of P2C is to **reduce annotation time**. In P2C, we employ Disco-occurrence Partition (DP) and group-wise prompting to remove a large number of irrelevant labels while maintaining **high recall** (see *Figure 4*). Since this stage uses multi-option prompting, each image requires only **one inference** (or ***l* inferences** if divided into *l* groups), which makes the inference process **much faster than binary prompting**.
> >
> > After the P2C stage, the number of candidate labels is significantly reduced, typically **fewer than one tenth of the total label space**. This reduction also leads to a substantial decrease in the number of inferences required in the second stage (CAD), thereby **greatly shortening the overall annotation time**. For example, on the COCO 2014 dataset, when using the LVLMAnt 7B model, the annotation time for **P2C** is **less than 4 h** , and for **CAD** is **less than 12 h**.
> >
> > **Q6**: In Table 3, there lacks a baseline that P2C + binary prompting, which makes the efficiency of CAD unclear.
> >
> > **R6**:  Thank you for the suggestion. We have added the **results of P2C + binary prompting** to *Table 3* (second row). Compared with P2C alone, incorporating binary prompting–based verification leads to a **significant improvement in annotation performance**. This result supports our **claim in the Introduction (line 70)** that using multi-option prompting to annotate a smaller candidate set **effectively reduces annotation cost**, while applying binary prompting for verification significantly **enhances annotation quality**.
> >
> > This structured annotation framework not only significantly reduces annotation cost but also notably improves annotation quality. This improvement can be clearly observed in comparison with the baseline Qwen2.5-VL (only using binary prompting), where P2C+binary prompting achieves substantial performance gains. During the P2C stage, the model **successfully identifies a large number of irrelevant labels**, some of which are difficult to detect using binary prompting alone. This advantage arises from the **complementary nature** of **multi-option prompting** and **binary prompting**, as the two annotation strategies exhibit **different preferences**. Consequently, their combination allows the model to **detect and filter out more irrelevant labels**, leading to a more accurate and efficient annotation process.
> >
> > Table 3: Ablation Studies on O365. S.-C. denotes super-category.
> >
> > | P2C  | Binary | S.-C. | Similarity | Description | OP    | OR    | OF1   | CP    | CR    | CF1   | mAP   |
> > | :--: | :----: | :---: | :--------: | :---------: | ----- | ----- | ----- | ----- | ----- | ----- | ----- |
> > |  ✔   |        |       |            |             | 28.27 | 84.87 | 42.41 | 18.54 | 87.06 | 27.66 | 39.63 |
> > |  ✔   |   ✔    |       |            |             | 54.99 | 70.21 | 61.67 | 43.64 | 70.39 | 53.88 | 45.84 |
> > |  ✔   |   ✔    |   ✔   |            |             | 56.22 | 72.12 | 63.19 | 42.24 | 71.68 | 53.15 | 46.08 |
> > |  ✔   |   ✔    |   ✔   |     ✔      |             | 56.72 | 70.82 | 62.99 | 43.15 | 71.12 | 53.71 | 46.29 |
> > |  ✔   |   ✔    |   ✔   |     ✔      |      ✔      | 60.43 | 68.52 | 64.22 | 46.99 | 67.79 | 55.50 | 46.99 |
> >
> > ####

---

> > > ### Author Response · Authors · 2025-11-28
> > >
> > > Dear HCdA,
> > >
> > > Thank you again for taking the time to provide such insightful and constructive comments.
> > > We hope that our rebuttal addressed the main concerns you raised.
> > > If any of our explanations require further detail or if additional points should be discussed, we would be glad to elaborate during the ongoing discussion phase.
> > >
> > > Best,
> > > The Authors

---

### Official Review · Reviewer_Nydn · 2025-11-01

**Soundness:** 3
**Presentation:** 3
**Contribution:** 3
**Rating:** 6
**Confidence:** 3

**Summary:**

The paper approaches a problem of evaluating whether a LVLM can replace human on the task of annotations. The study shows that on the common categories the LVLM outperforms humans than the ambiguous ones. After this they propose LVLMANT, a two-stage annotation framework. The first stage, Prompts-to-Candidates (P2C) generates a high recall candidate set and the second stage, Concept-Aligned Disambiguation (CAD), uses an LLM to automatically identify ambiguous category increases the precision. The results show that the annotation quality reaches human level by reducing costs by 36x.

**Strengths:**

1) The paper addresses an expensive bottleneck of manual data annotation in machine learning. This paper solves this in a systematic manner.

2) The paper achieves a very high 36x reduction in human annotation cost, hence proving to be a practical solution.

3) The two stage process ensures high recall in the first step and high precision in the next one. Overall aims to optimize for maximum performance.

**Weaknesses:**

1) The paper goes beyond the 9 page limit of ICLR. If it was not desk rejected , the paper seemed to have passed the initial stage though.

2) Although the human cost is reduced by 36x , the improvement on mAP is low.

3) The paper combines the existing techniques in two stages, so the technical contribution can be seen as shallow.

4) The VLLMs can make same hallucinations in the two stage and hence the 'noise' can amplify in subsequent stages.

**Questions:**

1) How does the P2C ensemble handle correlated hallucinations, the models can make same hallucinations that can propagate?

2) Given the two-stage framework, how to justify 1.34% mAP improvement on COCO is not just statistical noise?

3) How can the paper claim CAD is a novel technical contribution when it is based on closed source model that is ChatGPT-4o?

4) If human annotations are imperfect, are these used as ground truth?

---

> ### Author Response · Authors · 2025-11-21
>
> Thanks for your appreciation of our paper. We are glad that you considered our work “systematic, practical”. We are glad to answer all your questions.
>
> **Q1**: Although the human cost is reduced by 36x , the improvement on mAP is low.
>
> **R1**: In our approach, *Human-Assisted Calibration (HAC)* is an optional component.  Without using HAC (at zero manual annotation cost), our method LVLMAnt 7B outperforms the baseline (Qwen2.5-VL 7B) by **1.93% mAP** on **COCO 2014** and **2.44% mAP** on **O365**. When HAC is employed, LVLMAnt 7B + HAC achieves a **mAP of 82.84**, with an average of only **2.96 candidate labels per image pending annotation**. This represents an improvement of **+2.08 mAP** over the baseline (Qwen2.5-VL 7B) on COCO 2014. Considering the potential improvement range of **83.26 − 80.76 = 2.5** on COCO 2014, our proposed method in fact achieves a **substantial performance gain**. With a **36× reduction in human annotation cost**, our method shows a **performance gap of only 0.42 mAP** compared with full manual annotation.
>
> **Q2**: The paper combines the existing techniques in two stages, so the technical contribution can be seen as shallow.
>
> **R2**:  Our main contributions lie in two aspects: (1) a **systematic analysis** of LVLM-generated annotations for the image tagging task, and (2) the proposal of an LVLM annotation framework that improves annotation performance in terms of both **efficiency** and **effectiveness**. From a technical perspective, the two stages of the proposed method are **complementary**. The first stage (*Prompt-to-Candidate, P2C*) employs multi-option prompting, which **substantially reduces inference frequency (only one inference per image)** while producing a compact candidate set, typically containing fewer than one tenth of all categories. The second stage then utilizes binary prompting to carefully refine these candidate label sets, thereby **significantly enhancing annotation quality**.
>
> To fully leverage the strengths of both stages, we introduce two key techniques. In the **first stage**, we propose Divide-and-Conquer Prompting, which partitions the category space into multiple groups using a disco-occurrence partition strategy. We then perform group-wise prompting and fuse the inference results from all groups. This design **substantially improves annotation recall** (see *Figure 4*) while **adding almost no extra candidate labels**.
>
> In the **second stage**, we introduce Concept-Aligned Disambiguation (CAD), which addresses the **misalignment between category names and their intended semantic concepts**，including issues such as sense ambiguity, hypernym overreach, and misnomers. This module **significantly improves annotation precision** (see *Table 3*).
>
> To the best of our knowledge, **both of these techniques are proposed for the first time in our work**.
>
> **Q3**: The VLLMs can make same hallucinations in the two stage and hence the 'noise' can amplify in subsequent stages.
>
> **R3**: In our method, **hallucinations primarily occur in the first stage (P2C)**, while the second stage (**CAD**) is specifically designed to **eliminate** them. The goal of P2C is to generate a **candidate label set** that removes a large number of irrelevant labels while retaining as many true labels as possible. Our experiments demonstrate that the proposed P2C method effectively achieves this goal, ensuring **high recall**, although it may introduce some **noisy labels** (see *Figure 4*).
>
> The **CAD** stage is then responsible for **detecting and removing hallucinated labels** within the candidate set. It does **not introduce additional noise**; at worst, it may fail to identify some noisy labels. As shown in *Table 3*, CAD **effectively detects hallucinations** and filters out noisy labels, leading to a **substantial improvement in precision**, from **28.27%** in P2C to **60.43%** after applying CAD.
>
> **Q4**: How does the P2C ensemble handle correlated hallucinations, the models can make same hallucinations that can propagate?
>
> **R4**: The P2C ensemble is designed to further **improve recall**, ensuring that more true labels are included in the candidate set. Therefore, whether different models produce the same hallucinations does not affect the final results. As mentioned in R3, our proposed **CAD strategy** is applied to **eliminate hallucinated labels** within the candidate set and **enhance annotation precision**, which is verified by the experimental results presented in *Table 3*.

---

> > ### Author Response · Authors · 2025-11-21
> >
> > **Q5**: Given the two-stage framework, how to justify 1.34% mAP improvement on COCO is not just statistical noise?
> >
> > **R5**: The two stages together provide strong evidence for the overall effectiveness of our method. As shown in *Table 3*, the first stage (P2C) achieves an mAP of 39.63%, and the performance further increases to 46.99% after applying CAD for disambiguation. Across different datasets, the proposed LVLMAnt consistently demonstrates significant improvements. On COCO 2014, compared with the baseline, the LVLMAnt 7B and 32B models achieve gains of **1.93%** and **1.34%**, respectively. On O365, the improvements are **2.44%** and **1.99%**, respectively. These experimental results clearly indicate that the proposed method yields **consistent and significant performance improvements**. From a **numerical perspective**, mAP represents the average of AP across all categories. An improvement of 1.34% mAP therefore indicates that the **average AP per category increases by 1.34%** (COCO contains 80 categories), which makes it highly unlikely that the improvement is due to statistical noise.
> >
> > **Q6**: How can the paper claim CAD is a novel technical contribution when it is based on closed source model that is ChatGPT-4o?
> >
> > **R6**：The main contribution of CAD lies in its ability to **reduce the misalignment between category names and intended semantic concepts** through **automated prompt design**, which lowers the risk of hallucination, identifies noisy labels within the candidate set, and **significantly improves annotation precision**. In this process, ChatGPT-4o's task is to answer a series of questions to resolve the ambiguity in the annotation prompts. We can adopt any chatbot to perform this task. To ensure **reproducibility** and **fair comparison**, we employ ChatGPT-4o to fulfill this role, which does **not diminish the core contribution** of CAD.
> >
> > **Q7**: If human annotations are imperfect, are these used as ground truth?
> >
> > **R7**：Although human annotation is not perfect, widely adopted benchmarks such as COCO 2014 and O365 contain only a **very small proportion of noisy labels**. For example, in the COCO dataset, to minimize annotation noise, each image was labeled by **seven annotators**, although a small amount of noise is still inevitable. In our experiments, it can be observed that **human annotation further improves annotation quality**. As shown in *Table 2*, by incorporating human annotation, LVLMAnt + HAC achieves an improvement of **1.38% mAP** compared with LVLMAnt.

---

> > > ### Author Response · Authors · 2025-11-28
> > >
> > > Dear Nydn,
> > >
> > > Thank you again for taking the time to provide such insightful and constructive comments. We hope that our rebuttal addressed the main concerns you raised. If any of our explanations require further detail or if additional points should be discussed, we would be glad to elaborate during the ongoing discussion phase.
> > >
> > > Best,
> > > The Authors

---

### Meta-Review · Area_Chair_2GvU · 2026-01-07

**Summary:**

This paper addresses the problem of automating annotation for image tagging by leveraging large vision-language models for generating labels. The three reviewers pointed out several critical concerns about different aspects of this paper, including but not limited to:

1. The 9th page of this paper has content besides references, which violates the basic page limit restriction. It should be desk rejected.
2. The technical contribution combines the existing techniques in two stages is limited.
3. Comparison with other works are missing, such as CaSED and CLIP. Some experimental details are missing.
4. The VLLMs can make same hallucinations in the two stage and hence the 'noise' can amplify in subsequent stages.
5. Despite of reduced human cost, the improvement on mAP is low.
6. Some discussion on the experimental results are insufficient, such as the discussion on cases where LVLMANT fails, the choice of the sets of co-occurring and disco-occurring.
7. The robustness is not well demonstrated. Testing with weaker LVLMs (like LLava or InstructBLIP) would better demonstrate robustness, particularly whether CAD can handle noisier candidate sets from lower-capability models.

The current version needs a thorough revision and lies below the acceptance bar of ICLR. I hope the detailed reviews are helpful for the authors to improve the quality of this paper.

**Reviewer Concerns:**

The 3rd, 6th, and 7th concerns are partially addressed, while the others are still outstanding.

**Reviewer Scores:**

None.

---

### Decision · Program_Chairs · 2026-01-26

Reject